# Predictors of CIN2+ in Patients with PAP III-P (ASC-H): A Cross-Sectional Study

**DOI:** 10.3390/diagnostics13061066

**Published:** 2023-03-11

**Authors:** Luana Antonia Kaiser, Tomas Kupec, Laila Najjari, Elmar Stickeler, Julia Wittenborn

**Affiliations:** Department of Gynecology and Obstetrics, University Hospital of the RWTH Aachen, Pauwelsstrasse 30, 52074 Aachen, Germany

**Keywords:** transformation zone, colposcopy, cytology, HPV

## Abstract

Background: This study aims to investigate whether specific characteristics of the patient group with ASC-H (PAP III-p) findings increase the likelihood of clinically significant disease (CIN2+), offering implications for risk-adapted clinical management. Methods: 225 patients with an ASC-H smear presenting to our colposcopy unit between 2014 and 2021 were identified and examined retrospectively. A total of 203 patients were included in the cross-sectional study using multivariate logistic regression. Results: Of the 12 variables that entered the regression model, transformation zone (*p* = 0.045) and HPV infection (*p* = 0.005) contributed significantly to predicting CIN2+. A T3 transformation zone had a protective effect, reducing the likelihood of significant pathology. Infection with HPV high risk (non 16/18) or HPV high risk (16/18), on the contrary, increased the likelihood of CIN2+ four times and seven times, respectively, compared to the lower-risk category. Conclusion: A differential approach in the workup of ASC-H cytology is advisable. Younger, premenopausal patients with positive HPV high-risk findings are at a significantly higher risk for CIN2+ lesions than older postmenopausal women with negative high-risk HPV test results. As the risk increases from HPV high-risk non 16/18 to HPV 16/18 infections, using HPV PCR testing is advisable.

## 1. Introduction

Infections with high-risk human papillomaviruses (HPVs) are the leading risk factor for the development of cervical cancer and its precursor lesions, cervical intraepithelial neoplasia (CIN) [1,2,3,4]. The incidence of cervical cancer can be reduced by introducing organized screening programs to diagnose pre-invasive lesions [5,6,7,8,9]. In Germany, a newly organized screening based on cytology (PAP smear) and HPV testing was introduced in 2020, replacing the previous cytology-only based program [10]. Although cervical cancer is by now a rare disease in Germany, with approximately 4600 diagnosed cases per year, the number of CINs identified by screening is significantly higher [11,12].

In July 2014, the Munich Nomenclature III for gynecological cytodiagnosis of the cervix became effective in Germany [13]. For the first time, this revised nomenclature allowed the universal transferability of German study results by corresponding with the internationally established Bethesda system [13]. The new subcategories for unclear cytologic findings presented in the Munich Nomenclature III made these interpretations statistically detectable in Germany [13]. Its subcategory III-p corresponds to the “Atypical squamous cells, cannot exclude a high-grade squamous intraepithelial lesion” (ASC-H) category in the Bethesda system [13,14]. It subsumes cytological changes that are suggestive of a high-grade squamous intraepithelial lesion (HSIL), yet not sufficient for a definitive HSIL interpretation [13,14]. ASC-H, respectively, PAP III-p, is an uncommon subcategory only assigned to 0.11% of PAP findings in 2015 [11].

The German screening guidelines for abnormal findings demand a colposcopy within three months for all patients with an ASC-H screening result, regardless of the age group and HPV test result [15]. However, according to the German S3 guideline “Prevention of cervical cancer”, a further workup with additive methods (HR-HPV test or p16/ki-67 immunocytochemistry) is initially permissible within three months [16]. According to the German national guideline, only if the HR-HPV test or p16/ki-67 detection is positive, is colposcopy within three months indicated. In the case of negative test results, control by cytology and HPV test after 12 months is sufficient [16]. Thus, there is a discrepancy concerning the management of ASC-H in the screening algorithm of the federal health ministry and the national guidelines, showing the uncertainties of the correct management of this rare cytological finding.

ASC-H findings display a heterogeneous spectrum of underlying entities, ranging from benign findings to CIN3 and cervical cancer [11,17]. At the same time, a notable risk of clinically significant disease, defined as CIN2, CIN3, or cervical cancer, has been described for ASC-H interpretations [17,18]. However, the heterogeneity addressed leads to uncertainty among treating physicians and patients regarding the clinical significance of an ASC-H diagnosis. This study aims to investigate whether specific characteristics of the patient group with ASC-H findings increase the likelihood of clinically significant disease, offering implications for risk-adapted clinical management. Data were collected retrospectively, intending to characterize the patient population with ASC-H findings in our DGK-certified colposcopy unit.

## 2. Materials and Methods

### 2.1. Study Design and Population

From a total of 3791 patients examined at the dysplasia unit of the Department of Obstetrics and Gynecology at the University Hospital Aachen from July 2014 to August 2021, the 3218 patients presenting with cytological abnormalities were screened for a history of ASC-H or an ASC-H result at the time of examination. Patients who were presenting for vulvar or vaginal disorders were not considered for screening. Subsequently, all 225 identified patients with a documented ASC-H smear were examined retrospectively. Afterwards, patients who had a total hysterectomy with an ASC-H smear of the vaginal stump were excluded. Additional exclusion criteria were pelvic radiotherapy in temporal relation to an ASC-H PAP smear collection, a presentation at our dysplasia unit more than 12 months after an ASC-H result, and missing outcome data. Finally, 203 patients were included in the cross-sectional study (Figure 1).

During the time of investigation, 2374 high-grade lesions were diagnosed at our center of colposcopy. Of these, 1330 were excised.

### 2.2. Examination Method

The standardized colposcopic examination was performed in the DGK-certified colposcopy unit of the University Hospital Aachen by experienced and highly qualified AG CPC-certified personnel. It was executed with a Leisegang 3MCV colposcope. Every colposcopy included the systematic collection of a conventional PAP smear (control cytology). At the beginning of the study period, a test for human papillomavirus (HPV) was only taken if no external HPV test result was available or if it was older than six months at presentation. The collection of an HPV test at every colposcopy was standardized at a later stage during the study. The Seegene Anyplex II HPV 28 detection kit, an HPV DNA PCR test, was utilized throughout the study. This test kit identifies 19 high- and intermediate-risk HPV genotypes and 9 low-risk types.

In some cases, an additional swab was taken for microbiological examination at the examiner’s discretion. Indications were evidence of genital infection in the examination, such as an abnormal PH level; conspicuous fluor vaginalis; reddening of the vaginal wall; or reported clinical symptoms, such as itching, burning, or suspicious genital discharge. The colposcopic findings were classified according to the terminology of the International Federation for Cervical Pathology and Colposcopy (IFCPC) of 2011 for the cervix uteri [19]. The transformation zone was defined as 1, 2, or 3. Colposcopy was classified as “unsatisfactory” in patients with a non-examinable type 3 transformation zone, even with spreading of the cervical canal. This classification was selected to distinguish this finding from colposcopy that is “inadequate” for other reasons, such as a scarred cervix or bleeding. Subsequently, 5% acetic acid was applied to the cervical surface for further assessment. According to the IFCPC nomenclature, colposcopic findings were graded as “normal findings”, “minor change”, “major change”, and “suspicious of invasion/cancer”. For example, polyps, viral warts, or metaplasia were classified as “normal findings”. Thin, acetowhite epithelium, an irregular geographic border, a fine mosaic, and a fine punctation represented “minor changes”. The category of “major changes” included dense acetowhite epithelium, rapid appearance of acetowhitening, cuffed crypt (gland) openings, a coarse mosaic, coarse punctuation, a sharp border, an inner border sign, and a ridge sign. The term “suspicious for invasion/cancer” subsumed atypical vessels, fragile vessels, an irregular surface, exophytic lesions, necrosis, ulceration, and a visible tumor or gross neoplasm [19]. Under colposcopic control, a biopsy was taken from the most suspicious lesion area. In the case of multilocular lesions, several biopsies were taken. An additional endocervical curettage was conducted in patients with a type 3 transformation zone. The indication for surgical therapy, the loop electrosurgical excision procedure (LEEP), or total laparoscopic hysterectomy (TLH) was set according to the German S3 guideline for the prevention of cervical cancer [16]. The histopathological material was examined at the Institute of Pathology of the University Hospital Aachen. In women up to 24 years of age with a histopathologically proven CIN2 or CIN3 lesion, conservative management is performed according to the S3 guideline [16].

### 2.3. Data Collection

The patient’s baseline characteristics (Table 1) were recorded, including known risk factors for cervical cancer, e.g., HPV infection, immunosuppression, smoking, and taking of oral contraceptives [1,4,20,21,22,23,24,25,26,27]. The study aimed to identify predictors of significant pathology in cervical cancer screening patients with ASC-H smear.

“Significant pathology” was defined as the presence of CIN2, CIN3, adenocarcinoma in situ (AIS), or cervical cancer, respectively, CIN2+, in the histopathological findings of the biopsy, LEEP, or TLH. The category “non-significant pathology” subsumes CIN1 and benign findings of the categories “healthy”, “inflammation”, and “atrophy”. If the biopsy displayed a higher-grade CIN than the subsequent LEEP or TLH finding, it was assumed that the higher-grade lesion had already been completely removed during the biopsy. The results were considered consistent and classified according to the higher-grade finding. All data on the patients’ histopathological findings, as well as the classification into significant and non-significant pathology, are based on original diagnoses; no review of a histological specimen was performed. One patient was classified in the “non-significant pathology” group based on the visual diagnosis of extensive atrophy in the colposcopy; no histopathological sample was taken.

The patient’s menopausal status was gathered from previous medical findings or classified according to the German S3 Guideline “Peri- and Postmenopause- Diagnosis and Interventions” [28], based on the anamnestic information in the medical history form. Moreover, it was recorded whether the patients regularly received oral or vaginal hormonal replacement therapy. The identified HPV types were categorized according to the guidelines of the IACR (International Association of Cancer Registries) [29]. In the case of simultaneous infection with HPV types of different risk categories, the classification was based on the highest category. The detection of more than one HPV high-risk virus was defined as HPV high-risk multiple infections. “ASC-H in Domo” records whether the ASC-H was collected during the patient’s examination in our department or externally before the patient’s presentation to our department. A “Genital infection in temporal relation to ASC-H” was defined as a reported infection or bacterial vaginosis related to the external ASC-H. Pathogen detection in the external ASC-H smear or the control cytology in Domo or a positive test result of the simultaneously collected smear for microbiological examination was rated accordingly. A “history of HPV positivity” was determined as a positive HPV test result prior to the referral HPV result. A positive “history of abnormal cytology” was characterized as an abnormal PAP result, PAP II-p or ASC-US, and higher-grade findings, prior to the referral PAP smear. The variable “history of cervical conization” detects whether the patient received a LEEP or cold knife conization prior to the examination in our department.

### 2.4. Statistical Analysis

All statistical analyses were performed using IBM SPSS Statistics 27. Continuous variables are expressed as median ± interquartile range (IQR) and mean ± standard deviation (SD). The continuous variables were checked for normal distribution graphically utilizing a histogram with a normal distribution curve. Categorical variables are given as absolute frequencies and percentages.

Significant pathology (CIN2+) is regarded as the primary endpoint for the primary analysis. A frequency analysis of the histopathological outcome was conducted. Binomial logistic regression was performed to identify predictor variables of significant cervical pathology in patients with ASC-H smears. A crosstab between the categorical predictors and the outcome variable was created to identify empty or small cells with very few cases. Accordingly, categories of the variables “HPV infection” and “contraceptive method” were combined to prevent the logistic regression model from becoming unstable (Table 2). Since it does not represent the ordinal order of the variable “HPV infection”, the HPV category “high-risk positive (non-specified)” was coded as missing for the regression analyses. We chose not to include variables in the regression model that represent a different version of the endpoint (significant pathology/non-significant pathology) or a different method of measuring the endpoint. The variables “histopathological result of biopsy” and “histopathological result of LEEP or TLH” are different versions of the outcome variable. The variables “control cytology” and “result of colposcopic examination” are other diagnostic procedures, respectively, measurement methods used to diagnose dysplasia. The aim of the regression analysis was to identify clinical predictors of significant pathology, the investigation of a directional relationship between intrinsic patient characteristics and outcome.

Initially, all variables that do not constitute a different version of the outcome variable or a different measurement method to determine the outcome were tested univariately (Table 2). Variables with *p* < 0.2 were considered as candidates for the multivariate logistic regression model. The first category was chosen as the reference category for all categorical predictor variables in the dummy coding. For the final multivariate logistic regression model (Table 3), *p*-values < 0.05 were considered statistically significant.

## 3. Results

The patients’ baseline characteristics are summarized in Table 1.

**Table 1 diagnostics-13-01066-t001:** Baseline characteristics.

		Median	IQR	Mean	SD	Missing	Number	Number (%)
age (years)		43	19	44	13	0		
height (cm) *		166	9	167	7	5		
weight (kg)		68	20	71	16	6		
BMI ^a^		24.2	6.3	25.6	5.3	6		
menopausal status	premenopausal						121	59.6%
	perimenopausal						23	11.3%
	postmenopausal						57	28.1%
	missing						2	1.0%
contraceptive method	condom/no contraception						121	59.6%
	oral contraceptive						58	28.6%
	intrauterine device						16	7.9%
	three-month contraceptive injection						2	1.0%
	vaginal ring						2	1.0%
	missing						4	2.0%
hormone replacement therapy	no						195	96.1%
	yes						6	3.0%
	missing						2	1.0%
parity		1	2	1	1	0		
current pregnancy	no						189	93.1%
	yes						14	6.9%
smoking	never smoker						103	50.7%
	former smoker and currently smoking						97	47.8%
	missing						3	1.5%
family history of cervical cancer	no						155	76.4%
	yes						7	3.4%
	missing						41	20.2%
history of abnormal cytology	no						96	47.3%
	yes						104	51.2%
	missing						3	1.5%
history of cervical conization	no						184	90.6%
	yes						19	9.4%
Pap III-p (ASC-H) in domo	no						165	81.3%
	yes						38	18.7%
control cytology	II-a (NILM)						34	16.7%
	II-p (ASC-US)						50	24.6%
	II-g (AGC endocervical NOS)						2	1.0%
	II-e (Endometrial cells)						0	0.0%
	III-p (ASC-H)						38	18.7%
	III-g (AGC endocervical favor neoplastic)						0	0.0%
	III-e (AGC endometrial)						1	0.5%
	III-x (AGC favor neoplastic)						1	0.5%
	IIID1 (LSIL)						22	10.8%
	IIID2 (HSIL)						26	12.8%
	IVa-p (HSIL)						25	12.3%
	IVa-g (AIS)						0	0.0%
	IVb-p (HSIL with features suspicious for invasion)						2	1.0%
	IVb-g (AIS with features suspicious for invasion)						0	0.0%
	V-p (Squamous cell carcinoma)						0	0.0%
	V-g (Endocervical adenocarcinoma)						1	0.5%
	missing						1	0.5%
transformation zone	T1						78	38.4%
	T2						48	23.6%
	T3						74	36.5%
	missing						3	1.5%
result of the colposcopic examination	normal						12	5.9%
	minor change						42	20.7%
	major change						93	45.8%
	unsatisfactory						51	25.1%
	suspicious for invasion/cancer						4	2.0%
	missing						1	0.5%
histopathological result of biopsy	no dysplasia						77	37.9%
	CIN I						27	13.3%
	CIN II						27	13.3%
	CIN III						67	33.0%
	carcinoma						2	1.0%
	missing						3	1.5%
immunohistochemical examination of biopsy ^b^	no						118	59.0%
	yes						82	41.0%
recommendation of LEEP	no						101	49.8%
	yes						102	50.2%
histopathological result of LEEP or TLH	no dysplasia						15	14.7%
	CIN I						6	5.9%
	CIN II						11	10.8%
	CIN III						56	54.9%
	AIS						1	1.0%
	carcinoma						4	3.9%
	missing						9	8.8%
history of HPV positivity	no						130	64.0%
	yes						59	29.1%
	missing						14	6.9%
HPV vaccination	no						77	37.9%
	yes						4	2.0%
	missing						122	60.1%
HPV infection	negative						23	11.3%
	low-risk positive						3	1.5%
	intermediate-risk positive						9	4.4%
	high-risk positive (non 16/18)						76	37.4%
	high-risk positive (16/18)						78	38.4%
	high-risk negative						1	0.5%
	high-risk positive (non-specified)						5	2.5%
	missing						8	3.9%
HPV high-risk multiple infections	no						143	70.4%
	yes						47	23.2%
	missing						13	6.4%
genital infection in temporal relation to ASC-H	no						150	73.9%
	yes						53	26.1%
immunosuppression	no						192	94.6%
	yes						9	4.4%
	missing						2	1.0%
autoimmune disease	no						192	94.6%
	yes						11	5.4%
	missing						0	0.0%
previous or current malignant disease	no						190	93.6%
	yes						13	6.4%
	missing						0	0.0%
thromboembolic events or thrombophilia	no						192	94.6%
	yes						11	5.4%
	missing						0	0.0%
thyroid dysfunction	no						148	72.9%
	yes						55	27.1%
	missing						0	0.0%
cardiovascular disease	no						191	94.1%
	yes						12	5.9%
	missing						0	0.0%
arterial hypertension	no						175	86.2%
	yes						28	13.8%
	missing						0	0.0%
mental disorder	no						182	89.7%
	yes						21	10.3%
	missing						0	0.0%
other chronic diseases ^c^	no						181	89.2%
	yes						22	10.8%
	missing						0	0.0%
minor abdominal surgery in healthy patients ^d^	no						128	63.1%
	yes						69	34.0%
	missing						6	3.0%
c-section	no						178	87.7%
	yes						25	12.3%

* Normal distribution, ^a.^ BMI = weight (kg):(height(cm))^2^, ^b.^ In patients with performed biopsy, ^c.^ Bronchial asthma, chronic bronchitis, chronic inflammatory bowel disease, diabetes mellitus type 2, sickle cell anemia, ^d.^ Minor abdominal surgery for benign underlying disease (appendectomy, hernia surgery, laparoscopic surgery, e.g., cyst enucleation, adnexectomy, sterilization).

A frequency analysis of the histopathological outcome findings showed that 35.96% of the ASC-H patients examined had an underlying benign histopathological finding of the categories healthy, atrophy, or inflammation (Figure 2). Furthermore, 13.30% of ASC-H patients had a CIN1 finding, resulting in a non-significant pathology outcome for 49.26% of ASC-H cases. In total, 50.73% of the patients showed a significant pathology outcome (CIN2+).

In the univariate regression analysis, 14 of the 31 variables tested were significant (*p* < 0.2) (Table 2).

**Table 2 diagnostics-13-01066-t002:** Univariate logistic regression.

		*p*
age		**<0.001**
height		**0.179**
weight		0.730
BMI		0.500
menopausal status	premenopausal	**<0.001**
	perimenopausal	
	postmenopausal	
contraceptive method	condom/no contraception	0.330
	oral contraceptive	
	intrauterine device	
	others ^1^	
hormone replacement therapy	no	0.384
	yes	
parity		0.271
current pregnancy	no	**0.120**
	yes	
smoking	never smoker	0.990
	former smoker and currently smoking	
family history of cervical cancer	no	0.226
	yes	
history of abnormal cytology	no	**0.024**
	yes	
history of cervical conization	no	**0.003**
	yes	
Pap III-p (ASC-H) in domo	no	**0.183**
	yes	
transformation zone	T1	**<0.001**
	T2	
	T3	
history of HPV positivity	no	**0.001**
	yes	
HPV vaccination	no	**0.062**
	yes	
HPV infection	lower risk ^2^	**<0.001**
	high-risk positive (non 16/18)	
	high-risk positive (16/18)	
HPV high-risk multiple infections	no	0.313
	yes	
genital infection in temporal relation to ASC-H	no	**0.013**
	yes	
immunosuppression	no	0.295
	yes	
autoimmune disease	no	0.719
	yes	
previous or current malignant disease	no	**0.052**
	yes	
thromboembolic events or thrombophilia	no	0.719
	yes	
thyroid dysfunction	no	**0.014**
	yes	
cardiovascular disease	no	0.223
	yes	
arterial hypertension	no	0.624
	yes	
mental disorder	no	0.874
	yes	
other chronic diseases	no	0.706
	yes	
minor abdominal surgery in healthy patients	no	0.416
	yes	
c-section	no	0.770
	yes	

Significant values are indicated in bold. ^1.^ The contraceptive method category “others” subsumes the categories “three-month contraceptive injection” and “vaginal ring”. ^2.^ The HPV category “lower risk” subsumes the HPV categories “negative”, “low-risk positive”, “intermediate-risk positive”, and “high-risk negative”.

The final binomial logistic regression model (Table 3) was statistically significant, χ^2^ (15) = 65.597, *p* < 0.001, resulting in a good amount of explained variance, as shown by Nagelkerke’s R^2^ = 0.427 > 0.26 [30].

**Table 3 diagnostics-13-01066-t003:** Final multivariate logistic regression model.

							95% C.I. for EXP(B)	
Step 1	B	S.E.	Wald	df	Sig.	Exp(B)	Lower	Upper
height (cm)	−0.015	0.029	0.272	1	0.602	0.985	0.932	1.042
transformation zone			6.184	2	**0.045**			
transformation zone(1)	−0.553	0.476	1.347	1	0.246	0.575	0.226	1.463
transformation zone(2)	−1.409	0.567	6.182	1	**0.013**	0.244	0.081	0.742
menopausal status			5.991	2	0.050			
menopausal status(1)	−1.734	0.708	5.991	1	0.014	0.177	0.044	0.708
menopausal status(2)	−0.348	0.580	0.360	1	0.548	0.706	0.227	2.200
Pap III-p (ASC-H) in domo(1)	0.201	0.510	0.155	1	0.694	1.222	0.450	3.319
history of abnormal cytology(1)	−0.408	0.431	0.897	1	0.344	0.665	0.286	1.547
HPV infection			10.471	2	**0.005**			
HPV infection(1)	1.400	0.603	5.384	1	**0.020**	4.053	1.243	13.219
HPV infection(2)	2.003	0.620	10.433	1	**0.001**	7.410	2.198	24.980
current pregnancy(1)	−0.412	0.749	0.303	1	0.582	0.662	0.153	2.872
genital infection in temporal relation to ASC-H(1)	−0.725	0.453	2.563	1	0.109	0.484	0.199	1.177
history of HPV positivity(1)	−0.561	0.473	1.407	1	0.236	0.571	0.226	1.442
history of cervical conization(1)	−1.375	1.169	1.383	1	0.240	0.253	0.026	2.500

Significant values are indicated in bold. Variable(s) entered in step 1: height (cm), transformation zone, menopausal status, Pap III-p (ASC-H) in domo, history of abnormal cytology, HPV infection, current pregnancy, genital infection in temporal relation to ASC-H, history of HPV positivity, history of cervical conization.

The overall percentage of accuracy in classification was 74.1%, with a sensitivity of 77.3% and a specificity of 70.7%. Of the 12 variables that entered the regression model, transformation zone (*p* = 0.045) and HPV infection (*p* = 0.005) contributed significantly to predicting CIN2+. Compared to the reference category of patients with a T1 transformation zone, T3 transformation zones had a protective effect, reducing the likelihood of significant pathology, OR = 0.244 (95%-CI [0.081, 0.742]). Infection with HPV high risk (non 16/18) or HPV high risk (16/18), on the contrary, increased the likelihood of CIN2+ compared to the lower-risk category, OR = 4.053 (95%-CI [1.243, 13.219]) and OR = 7.410 (95%-CI [2.198, 24.980]). The variable menopausal status (*p* = 0.050) showed a low significant effect. The model included 170 cases; 33 subjects were excluded from the analysis due to missing data.

The univariate significant variable “HPV vaccination” (*p* = 0.062, missing = 122) could not be included in the multivariate model due to a too high number of missing values. The variable “age” was excluded from the final multivariate regression model due to a lack of significance in the multivariate regression and the known correlation with the variables “menopausal status” and “transformation zone”.

A stepwise forward selection (conditional) was conducted. The established selection model (Table 4) was statistically significant, χ^2^ (4) = 58.478, *p* < 0.001, resulting in a good amount of explained variance, as shown by Nagelkerke’s R^2^ = 0.358 > 0.26 [30].

The final selection model consisted of the variables transformation zone, entered in step one, and HPV infection, entered in step two. At each step, variables were chosen based on *p*-values of F, using a *p*-value < 0.05 as the default entry criterion. The overall percentage of accuracy in classification was 74.3%, with a sensitivity of 77.9% and a specificity of 70.7%. A total of 16 subjects have been excluded from the analysis due to missing data; the final selection model included 187 cases.

## 4. Discussion

In the present study, we were able to show that high-risk human papillomavirus infection is a predictor of significant underlying pathology, a CIN2+ lesion, in ASC-H patients (Table 3 and Table 4). The likelihood of CIN2+ was 4 times higher for patients with an HPV high-risk non 16/18 infection than for patients in the lower risk category and even 7 times higher for patients with an HPV infection of type 16 or 18 (OR = 4.053, 7.410) (Table 3). These findings are consistent with the fact that HPV infection is the leading cause of cervical cancer [1,2,3,4], and they show that the HPV PCR diagnostic is a valuable tool in the collective of ASC-H patients. The discrimination between HPV high-risk non 16/18 and HPV 16/18 gives additional information on the risk status. The genital HPV types are classified according to their risk potential for inducing invasive cervical cancer [29]. Especially persistent infection with high-risk HPV types can lead over time to the development of cervical intraepithelial neoplasia (CIN) and eventually to the genesis of cervical cancer [2,3,31]. Overall, 60 to 71 percent of cervical cancer cases are attributed to high-risk HPV subtypes 16 and 18 [3,32,33].

According to the German screening algorithm for abnormal findings in cytology, a colposcopic examination is indicated within three months for all patients with ASC-H findings, regardless of age and HPV status [15]. However, the German S3 guideline “Prevention of cervical cancer” states that an initial workup through high-risk HPV testing within three months is permissible; colposcopy must only be performed if the HPV test is positive [16]. This recommendation for triage of ASC-H patients by HR HPV testing using the Hybrid Capture 2 assay (HC2) is based on the results of a meta-analysis, which showed a sensitivity of 94.8% and specificity of 38.5% for detecting CIN2+ in women with ASC-H [16]. Another meta-analysis on the triage of ASC-H by Xu et al., published in 2016, reported a pooled absolute sensitivity of 93% and specificity of 45% of HC2 for CIN2+ detection in ASC-H patients [18]. Moreover, Xu et al. reported that genotyping for HPV 16 and 18 increased specificity to 74%, but resulted in a loss of sensitivity compared to HC2 testing to identify underlying CIN2+ lesions in ASC-H patients [18]. Here, we present data of patients tested by HPV PCR. This described loss of sensitivity by HPV 16 and 18 genotyping corresponds to our finding that HPV high-risk types other than types 16 and 18 are also significant predictors of an underlying CIN2+ lesion and, thus, cannot be neglected. According to our analysis, infections with HPV type 16 or 18 are particularly highly associated with CIN2+ pathology in ASC-H patients; their presence is crucial information that can be gained by using HPV PCR testing instead of HC2. Our main findings regarding HPV high-risk positivity show the importance of HPV high-risk testing in managing patients with ASC-H [16,18]. An available HPV PCR test result helps clinicians to assess the likelihood of an underlying CIN2+ lesion and, thus, classify the potential significance of an ASC-H finding in affected patients.

As a second factor, the transformation zone proved to be a significant predictor of CIN2+ in patients with ASC-H in the multivariate model (Table 3) (*p* = 0.045) and the selection model (Table 4) (*p* < 0.001). In this context, a type 3 transformation zone was a protective factor, reducing the risk of significant pathology by a factor of 0.244 (Table 3) compared to patients with a T1 transformation zone. This result is consistent with a recently published study by Maffini et al., who confirmed a type 3 transformation zone finding as a significant protective factor [34]. We hypothesize that T3 transformation zones are associated with older and postmenopausal patients, in whom ASC-H results are more often associated with atrophic changes than in younger patients. Chen et al. also reported that most HR HPV-negative postmenopausal women in their studied ASC-H cohort had atrophy-related changes, which may have led to overdiagnosis [35]. Atrophic cells in PAP smears from postmenopausal women may mimic HSIL and lead to classification as ASC-H [14,36]. In the present study, approximately 36% of ASC-H patients had a benign outcome; this group’s second most common finding was atrophic change (Figure 2). Our observations support the notable role of atrophy in PAP smear interpretations of postmenopausal patients. Thus, atrophy is the most common benign finding in our postmenopausal patient group and occurs with similar frequency as CIN2+.

Several studies have identified age as an important factor in disease outcome in ASC-H patients, showing that younger women are more likely to display CIN2+ lesions than older patient groups [17,35,37,38,39,40]. Furthermore, it was demonstrated that in patients with ASC-H cytology, there is an age-dependent HR HPV positivity that is exceptionally high in younger patients and decreases with age [35,37,41]. In this context, additional HR HPV testing in women over 40 years of age with ASC-H cytology was considered particularly helpful in identifying significant dysplasia [35] and a reasonable option for triage [37]. In contradiction with the findings addressed above [17,35,37,38,39,40], patient age was not identified as a significant predictor of CIN2+ in our analysis, as described in the results.

Considering the association between age and menopausal status, premenopausal women with ASC-H are more likely to show significant pathology than postmenopausal women [17,35,37,38,39,40]. Few studies with small sample sizes examine menopausal status as an isolated influential factor with respect to disease outcome in ASC-H [34,38]. These described only a borderline correlation between premenopausal status and CIN2+ outcome [34] or no significant influence of menopausal status at all [38]. On the other hand, results from Saad et al. [42] and Selvaggi et al. [43] support the presumed relationship between menopausal status and significant pathology in ASC-H patients. However, their classification of menopausal status was based strictly on age and not on clinical information, which must be named as a substantial methodological weakness of their work [42,43]. In our multivariate analysis (Table 3), the effects related to menopausal status were only slightly significant (*p* = 0.050) and should only be understood as a trend. Perimenopausal status reduced the likelihood of CIN2+ in comparison to premenopausal status (*p* = 0.014), OR = 0.177 (95%-CI [0.044, 0.708]). Future studies with larger sample sizes might demonstrate corresponding effects. Nevertheless, we observed that CIN2+ was the most common outcome in the premenopausal patient group; premenopausal women were more likely to have significant pathology than those with peri- or postmenopausal status.

In this study, there appeared to be mixed results concerning age, menopausal status, and T3 transformation zone. However, the results suggest that the biological age of patients, expressed by the T3 transformation zone, plays a role in the likelihood of underlying significant pathology in ASC-H patients.

In our opinion, a differential approach in managing patients with ASC-H would be advisable. Younger, premenopausal patients with positive HPV high-risk findings could be evaluated by colposcopy. In older, postmenopausal patients with negative HPV high-risk findings, a wait-and-see approach seems reasonable. In this group of ASC-H patients, local estrogenization could be performed to prevent over-interpretation in cytomorphology due to atrophic changes, followed by a repeat smear [14]. Subsequently, follow-up by PAP smear and HPV high-risk testing at 12 months could be recommended [16].

When looking at the mere diagnostic accuracy in predicting significant pathology, of course much higher sensitivity and specificity values can be achieved when including the result of the colposcopy in the model. However, in the presented analysis, we sought intrinsic patient characteristics that are not examiner dependent for a better characterization of the patient collective. Thus, our results can help triaging patients for immediate colposcopy versus a wait-and-see strategy based on their intrinsic characteristics.

This is a retrospective study design with a specific patient population and a limited number of cases. Due to the retrospective approach, some limitations must be addressed. At the beginning of the study period, no new HPV smear was taken during the colposcopy if the external HPV smear was not older than six months at the time of presentation. Furthermore, the pathologist’s evaluation of the PAP smear could have been biased because the histological sample of the respective patient was simultaneously available for diagnosis. In addition, in some cases, not all anamnestic factors were documented, which led to missing data in the analysis. During the study period, the patient’s HPV vaccination status was not collected by default. However, ASC-H is a rare subcategory, and this study collected a variety of sociodemographic and clinical risk factors and included a wide range of patients. Previous research on predictors of clinically significant disease in ASC-H patients has frequently drawn on fewer cases or only looked at a small number of putative predictors [34,38]. The results presented here provide important information for future research in prospective studies with larger numbers of cases.

## 5. Conclusions

Our data show that HPV PCR testing with discrimination of HPV 16/18 and non-16/18 HR infections and consideration of biological age is significant for the risk assessment of underlying clinically significant disease (CIN2+) in patients with ASC-H findings. Younger, premenopausal patients with positive HPV high-risk findings are at a significantly higher risk for CIN2+ lesions than older postmenopausal women with negative high-risk HPV test results. Therefore, a differential approach in the workup of ASC-H cytology is advisable.

## Figures and Tables

**Figure 1 diagnostics-13-01066-f001:**
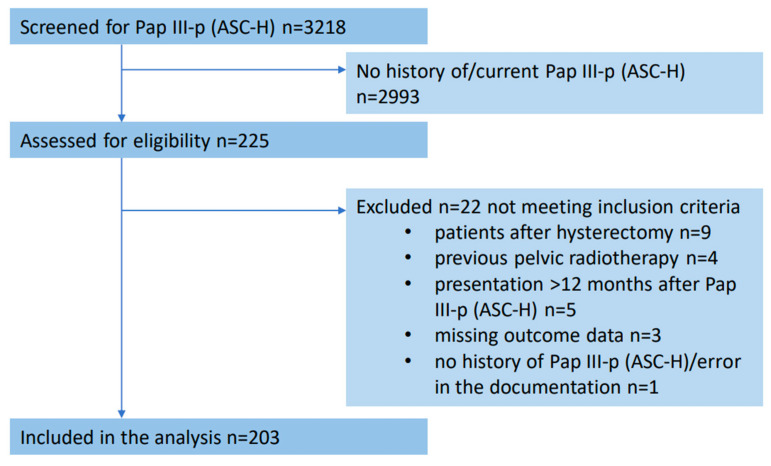
Patient Flow Diagram.

**Figure 2 diagnostics-13-01066-f002:**
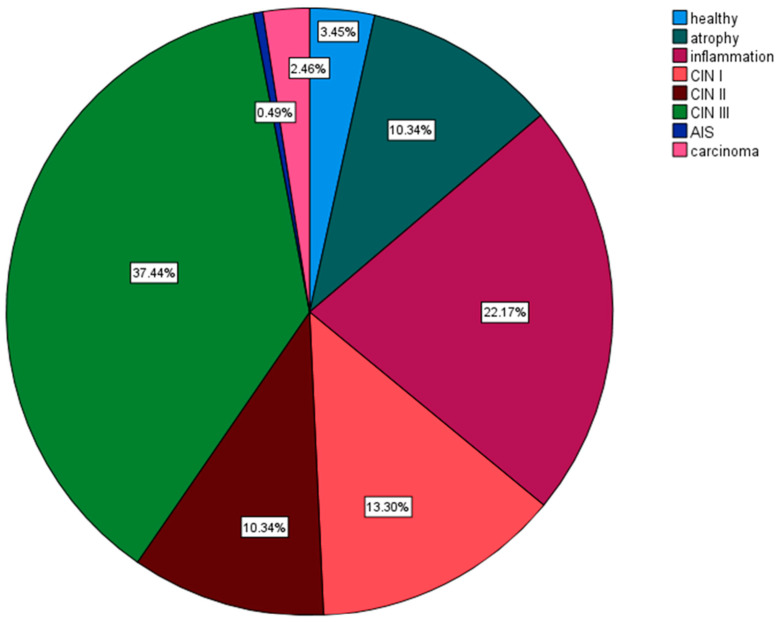
Histopathological outcome.

**Table 4 diagnostics-13-01066-t004:** Selection model.

								95% C.I. for EXP(B)	
		B	S.E.	Wald	df	Sig.	Exp(B)	Lower	Upper
Step 1 ^a^	transformation zone			37.200	2	0.000			
	transformation zone(1)	−1.148	0.410	7.850	1	0.005	0.317	0.142	0.708
	transformation zone(2)	−2.451	0.402	37.156	1	0.000	0.086	0.039	0.190
	constant	1.235	0.284	18.896	1	0.000	3.437		
Step 2 ^b^	transformation zone			31.523	2	0.000			
	transformation zone(1)	−0.990	0.430	5.292	1	0.021	0.372	0.160	0.864
	transformation zone(2)	−2.327	0.416	31.316	1	0.000	0.098	0.043	0.220
	HPV infection			12.085	2	0.002			
	HPV infection(1)	1.434	0.550	6.798	1	0.009	4.194	1.427	12.321
	HPV infection(2)	1.918	0.552	12.070	1	0.001	6.809	2.307	20.097
	constant	−0.267	0.540	0.244	1	0.621	0.766		

^a.^ Variable(s) entered in step 1: transformation zone, ^b.^ Variable(s) entered in step 2: HPV infection.

## Data Availability

The datasets generated during the current study are available from the corresponding author on reasonable request.

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
