# Peer review of "Predictors of CIN2+ in Patients with PAP III-P (ASC-H): A Cross-Sectional Study"

_diagnostics, 2023, doi:10.3390/diagnostics13061066_

Round 1

Reviewer 1 Report

I read with great interest the Manuscript titled "Predictors of CIN2+ in patients with PAP III-P (ASC-H): a cross-sectional study" which falls within the aim of the Journal.

In my honest opinion, the topic is interesting enough to attract the readers’ attention. 

Although the manuscript can be considered already of high quality, I would suggest to take into account the following minor recommendations:

- I suggest another round of language revision, in order to correct few typos and improve readability.

-The introduction should be extended and completed. I find interesting a reference to the efforts made for the prevention and early diagnosis of gynecological cancers (see PMID: 36141217).

- Although it is a retrospective analysis, inclusion/exclusion criteria should be better clarified by extending their description.

- Discussions can be expanded and improved by citing relevant articles (I suggest authors to read and insert in references the following article PMID: 35742340).

Reviewer 2 Report

This German retrospective study evaluated the risk of CIN2+ with ASC-H cytology. The methodology of this study is not clear. The authors do not indicate how many patients are referred to their centre for cytological abnormalities each year and what percentage of abnormal cytologic results an ASC-H category (only 0.11% in the literature, line 43) represents? It is not clear how many high-grade lesions are diagnosed in the authors’ institution each year and what is the percentage of CIN2 and CIN3 among these. Furthermore, it is not clear whether the same pathologist evaluated the PAP smear and available histological sample. Were histological samples reviewed or was original diagnosis included in the final analysis? Basic patient characteristics are included within the methods section and the authors do not mention a possible conservative management of high-risk lesions in young patients.

Reviewer 3 Report

Authors presented the statistical proof that infection with oncogenic HPV is a significant risk for CIN2+ pathology of the cervix. Obviously the conclusion is not novel but well documented. The model combining HPV infection and transformation zone 3 was able to predict CIN2+ pathology with sensitivity of 78% and specificity of 72%. The paper is well written, the study, although retrospective was planned properly, thus the conclusions seem to be supported by the results. The only concern is about the colposcopy. Why only the transition zone was implemented to the model? What about the rest of colposcopic features? Could authors probably compare the value of their model to the value of colposcopic examination?

Round 2

Reviewer 2 Report

I have no further comments

Reviewer 3 Report

Thank you for your precise answers. Adding these two explanations has made the text more comprehensive